# Changing Trends in Surgical Management of Nephrolithiasis among Young Adults: A 15-Year Population-Based Study

**DOI:** 10.3390/jpm12081345

**Published:** 2022-08-21

**Authors:** Dorit E. Zilberman, Tomer Erlich, Nir Kleinmann, Itay M. Sabler, Amos Neheman, Guy Verhovsky

**Affiliations:** 1Department of Urology, Chaim Sheba Medical Center, Tel Hashomer, Ramat Gan 52621, Israel; 2Department of Military Medicine, Hebrew University of Jerusalem, Jerusalem and the Israel Defense Forces Medical Corps, Ramat Gan 91905, Israel; 3Department of Urology, Sackler School of Medicine, Tel-Aviv University, Zerifin 60930, Israel; 4Shamir (Assaf Harofeh) Medical Center, Sackler School of Medicine, Tel-Aviv University, Tel-Aviv 69978, Israel

**Keywords:** nephrolithiasis, surgery, trends, young adults

## Abstract

**Background:** Increases in obesity and diabetes rates among all ages have led to a greater prevalence of nephrolithiasis worldwide. We aimed to explore the changing trends in surgical management of nephrolithiasis in young adults over a 15 year period. **Methods:** We reviewed medical records of military personnel for information on the diagnosis and care of nephrolithiasis before and during active service between 2007–2021, divided into three 5 year periods: 2007–2011, 2012–2016, and 2017–2021. Demographic, clinical, radiological, and surgical data were retrieved for the analysis of changing trends. **Results:** The records of 1,117,692 recruits yielded 7383 (0.66%) with stone-related surgeries, of whom 1885 were operated during military service. Their median age was 19.6 years (interquartile range [IQR] 16.8–21.2), 829 (70%) were males, and the cohort’s median body mass index was 23.6 (IQR 17.3–26.1). There was a dramatic decline in shock wave lithotripsy (SWL) prevalence (35.1%, 10.4%, and 4.4%, respectively) with a continually increasing prevalence of ureteroscopy (URS)/retrograde intrarenal surgery (RIRS) (62.7%, 88.5%, and 94.6%, *p* = 0.01). Percutaneous nephrolithotomy (PCNL) procedures have become nearly extinct over time (0.8% in 2017–2021). The number of median-sized stones treated by URS/RIRS increased (7.5 mm, 8.2 mm, and 9.7 mm, *p* = 0.044), but not those treated by SWL/PCNL. The median length of medical leave for URS/RIRS and PCNL decreased significantly (7 vs. 4 days, *p* = 0.05 and 10 vs. 6 days, *p* = 0.036, respectively), with no comparable change for SWL. There was a substantial decline in ancillary procedures in the URS/RIRS groups (9%, 6.8%, and 3.1%, *p* < 0.01), but not in the SWL/PCNL groups. **Conclusions:** Advancements in technology and surgical training are leading to the extinction of SWL and the adoption of URS/RIRS as the new standard of care for nephrolithiasis among young adults.

## 1. Introduction

Nephrolithiasis has become widespread over the past decade due to significant increases in obesity and diabetes [1,2]. The median age at diagnosis has remained stable at 49 years, as has the estimated lifetime prevalence of 13% [3]. Although the number of nephrolithiasis cases in the western world is consistently increasing over time [1,4,5], that increase has mostly affected the >59 year age group, while the rates for the 15–59 year age group have remained stable [3]. The annual prevalence for the 8–18 year age group has been estimated at 2.78 cases per 10,000 patients [6].

The reports in the literature on nephrolithiasis describe a variety of surgical approaches for the adult population as well as their changing popularity over time. A tremendous increase in percutaneous nephrolithotomy (PCNL) cases was documented in the United States between 1998 and 2013, accompanied by a concomitant increase in annual cases [2,7]. Trends towards moderate increases and even decreases were observed in Canada [8], Great Britain [3], and Australia [9]. Similarly, a recent study from the United States showed the beginnings of a decreasing trend in PCNL cases after reaching a plateau between 2008–2011 [10].

The popularity of shock wave lithotripsy (SWL) has consistently diminished over time worldwide, while there have been consistently-increasing cases of ureteroscopy (URS) [1,3,8,11,12,13]. This trend for SWL, however, was not observed in the pediatric population, for whom it has been and remains the most popular procedure for managing kidney stone disease [6].

The purpose of the present study was to explore the trends in surgical management of nephrolithiasis in a large population of young adults, who represent a population between children and adults that has not been studied in isolation.

## 2. Materials and Methods

### 2.1. Study Participants

Following approval by the Israel Defense Force Medical Corps Review Board (#2099–2020), we conducted a retrospective cohort study of all Israeli adolescents (~17 years of age) who were potentially eligible to join the military forces between 2007–2021. Their eligibility for military service is determined by a medical board examination, which is conducted at recruitment centers. A medical history from their primary care physician is reviewed on a structured form, a current medical history is obtained, and a physical examination is conducted. Recruits with a history of nephrolithiasis are referred to a board-certified urologist or nephrologist for confirmation of the diagnosis. A report from a primary care physician, urologist, or nephrologist serves to diagnose nephrolithiasis and provide its associated past history, including prior operations. A committee of two certified military service physicians assesses and verifies the correctness and completeness of each entry in the provided medical information. Each diagnosis is given a designated numerical code, assigned a level of severity, and entered into the central database. For inclusion in this study, participants had to be 16–19 years of age at the time of their medical board examination.

### 2.2. Data Collection and Definition of Variables

The medical records of our military forces supplied the information on the nephrolithiasis diagnosis and care during the recruit’s service, including demographic, clinical, radiological, as well as surgical and procedure data. Radiographic imaging, ideally non-contrast helical computed tomography (NCCT), was used to assess stone load prior to therapy. Procedure type (i.e., SWL, URS/retrograde intrarenal surgery [RIRS], PCNL), procedure dates, and number of procedures were included in the surgical/procedural data collected during military duty. Surgical procedure codes were retrieved from billing records and matched to individual participants. An additional stone surgery that took place at least 3 months after the first surgery was counted as a separate event. That 3 month interval aimed to exclude cases in which the first procedure was not a definitive one but rather only a ureteral stent insertion. The post-procedure recovery period (days) was also included in the clinical data. Ancillary procedures were defined as any additional surgeries that were carried out during the 3 month period following the primary surgery. Of note, surgeries have been carried out in national civilian hospitals across the country.

The cohort was divided into three 5 year time periods (2007–2011, 2012–2016, and 2017–2021) in order to identify trends in the management of nephrolithiasis among otherwise healthy young adults during the preceding 15 years.

### 2.3. Statistical Analysis

Continuous variables are presented as median (interquartile range [IQR]) and categorical variables as percentages of individuals in each group. Categorical variables were compared by chi-square or Fisher’s exact tests. Kruskal-Wallis was implemented for the comparison of multiple groups consisting of not-normally distributed variables. Coefficients are presented as 95% confidence intervals (95% CI) and *p*-values. URS and RIRS cases were analyzed as a single group (“URS/RIRS”) and compared to both the SWL and PCNL groups. *p* values ≤ 0.05 were considered significant. Statistical analyses were performed with IBM SPSS, version 23.0 (Armonk, NY, USA: IBM Corp., 2015).

## 3. Results

In total, 7383 of the 1,117,692 recruits (0.66%) had stone-related surgery, out of whom 1885 (25%) were operated on during their military service and comprised the current study group. Their median age was 19.6 years (16.8–21.2). Of those 829 (70%) were males, and their median body mass index was 23.6 (17.3–26.1). A total of 113 people (6%) had undergone a second stone surgery. The median follow-up period was 37 months (22–48).

Figure 1 depicts the changing trends in surgical management of nephrolithiasis during a 15 year period divided into three parts: 2007–2021, 2012–2016, and 2017–2021.

There was a dramatic decline in SWL prevalence, with a continuously increasing prevalence of URS/RIRS cases (*p* = 0.01). Cases of PCNL have reached near-extinction over time. There was no significant difference in the size of stones treated by PCNL or SWL, but there was a significant increase in the median stone size treated by URS/RIRS throughout the 15 year period (7.5 mm in 2007–2011 vs. 8.2 mm in 2007–2016 vs. 9.7 mm in 2017–2021, *p* = 0.044, Table 1).

The length of the postoperative medical leave from military duty is detailed in Table 2.

The median leave for URS/RIRS and PCNL decreased significantly during the 15 year study period (7 vs. 4 days, *p* = 0.05 and 10 vs. 6 days, *p* = 0.036, respectively), while there was no change for SWL. There was a substantial decline in ancillary procedures during the follow-up period in the URS/RIRS group (9% in 2007–2011, 6.8% in 2012–2016, and 3.1% in 2017–2022, *p* < 0.01), but not in the SWL or PCNL groups (Figure 2).

## 4. Discussion

The most recent American (2016) and European Urological Association (2022) guidelines on nephrolithiasis recommend that, similar to adults, children should be offered SWL or retrograde intrarenal surgery (RIRS) for stones <20 mm, while stones >20 mm will preferably go to PCNL [14,15].

Traditionally, SWL was the first line of treatment in children with nephrolithiasis as their small body mass, low fat content, and relatively small and soft stones have all led to higher success rates using this method [6].

Like the pediatric population with nephrolithiasis, young adults with nephrolithiasis are usually free of the associated comorbidities seen in adults and have similar relatively low body mass and fat content.

We therefore expected to observe trends among them similar to those observed by Park et al. among children [6]. The authors’ study included 701 children with a median age of 13 years (range 0–17) who were surgically treated for stones between 2007–2014. Those authors observed that SWL was the most common procedure, representing a total of 66% of all recorded cases. The yearly number of SWL cases has not changed throughout the years, while the number of URS/RIRS procedures for larger stones increased from 15% to 31% of the cases, and PCNL procedures decreased from 13% to fewer than 10% of the cases.

Opposite to our expectation, we observed a near-total disappearance of SWL. As has been seen for adult populations in recent studies that pointed to an overall decline in SWL rates from a total of 50–70% to 34–37% as of 2010 to the present [8,11,13], we observed a prominent steep decline from 35.1% between 2007–2011 to 4.4% between 2017–2021.

This dramatic disappearance can be partly explained by the aging of the first-generation HM3 Dornier lithotripter (Dornier Medizintechnik GmbH, Germering, Germany) and the encumbrance associated with its operation, i.e., its water bath and the requirement of regional or general anesthesia. So far, all attempts to produce lithotripters that are as effective as the HM3 Dornier but are more compact and user-friendly have failed [16]. Consequently, spare parts for the original HM3 lithotripter have become sparse and difficult to acquire, and medical centers have been reluctant to purchase newer-generation lithotripters. Parallel to this, the endoscopic arsenal has advanced dramatically, and there has been a significant increase in endourology-trained urologists, particularly among younger ones [11]. The latter have felt much more comfortable with endoscopic procedures, which has subsequently translated into a constantly increasing rate of endoscopic procedures (from 62.7% up to 95%), as well as a subsequent increase in the size of the stone mass treated endoscopically.

Although URS/RIRS has mainly replaced SWL cases, we also evidenced a significant decline in PCNL cases: they fell from 2.2% between 2007–2011 to 0.8% between 2017–2021, a decline that was probably attributed to the same reasons.

This decline in PCNL popularity observed by us in a young adult population was also shown in other studies on pediatric populations [6]. It was also reported for patients of 50 years of age and above, among whom the annual PCNL rate varied between 3–7% between 1991–2017 and gradually declined or remained unchanged during the past decade [3,8,10,11,12,13].

Similar to the findings of Ordon et al. [8], we observed a decline in the ancillary procedures over time, mainly in URS, due to much more effective stone fragmentation. We credit the shortened medical leave period among the recruits to both the gradual improvement in equipment quality as well as the trend towards greater improvement and success of the surgical procedures by the local urological community.

To the best of our knowledge, this is the first analysis of the changing trends in surgical management of nephrolithiasis to be conducted on a young adult population (median age of 19.6 years). Since military service in Israel is mandatory for both sexes, the military database is a fair reflection of the real-life status of young Israeli adults from the standpoint of medical status in general and of nephrolithiasis in particular. Our results support the 2016 prediction of Heers et al. that “we are likely to see ureteroscopy overtake SWL for the first time in the next few years” [3].

Our study is not without limitations. Its main weakness is that its population is concentrated in a very small geographical region and one that enjoys very homogenous and very advanced medical treatment. As such, the findings may not reliably reflect the conditions in other parts of the world. Future studies on the management of young adults with nephrolithiasis carried out elsewhere are required to validate the results of the present study.

## 5. Conclusions

As has been seen for children and adults, the advancements in technology and urological training have led to the disappearance of traditional technologies and the adoption of endoscopic surgery for the treatment of nephrolithiasis in young adults. We predict that endoscopic surgery may soon become the new standard of care for most stones and for all age groups.

## Figures and Tables

**Figure 1 jpm-12-01345-f001:**
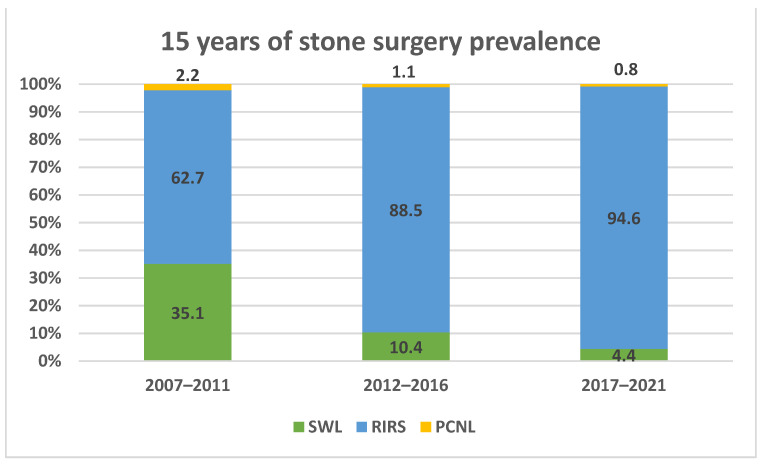
Stone surgery prevalence (%) after 15 years. SWL = shock wave lithotripsy; URS/RIRS = retrograde intrarenal surgery; PCNL = percutaneous nephrolithotomy.

**Figure 2 jpm-12-01345-f002:**
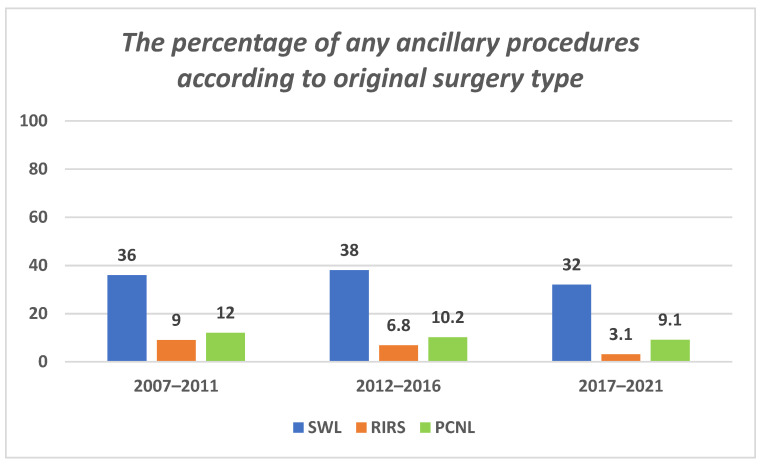
Any ancillary procedures according to the original surgery type.

**Table 1 jpm-12-01345-t001:** Stone size (mm, median [IQR]).

	2007–2011	2012–2016	2017–2021	*p*
SWL	4.1 (3.4–6.3)	5.1 (3.6–7.2)	4.9 (3.4–6.8)	0.61
URS/RIRS	7.5 (4.6–10.3)	8.2 (5.2–12.4)	9.7 (5.2–14.1)	0.044
PCNL	17 (14–22)	17 (12.6–23.3)	16.2 (13.8–21.1)	0.058

IQR = interquartile range; SWL = shock wave lithotripsy; URS/RIRS = ureteroscopy/retrograde intrarenal surgery; PCNL = percutaneous nephrolithotomy.

**Table 2 jpm-12-01345-t002:** Postoperative medical leave (days, median [IQR]).

	2007–2011	2012–2016	2017–2021	*p*
SWL	4 (1–7)	4 (1–6)	3(1–4)	0.42
URS/RIRS	7 (1–7)	4 (1–5)	4 (1–4)	0.05
PCNL	10 (3–14)	6 (3–11)	6 (3–7)	0.036

IQR = interquartile range; SWL = shock wave lithotripsy; URS/RIRS = retrograde intrarenal surgery; PCNL = percutaneous nephrolithotomy.

## Data Availability

Restrictions apply to the availability of these data. Data was obtained from the Israel Defense Force Medical Corps Medical records database and are available with the permission of the Department of Military Medicine, Hebrew University of Jerusalem, Israel.

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
