# Peer review of "Changing Trends in Surgical Management of Nephrolithiasis among Young Adults: A 15-Year Population-Based Study"

_jpm, 2022, doi:10.3390/jpm12081345_

Round 1

Reviewer 1 Report

I think that your article is an interesting study in that it examines the evolution of surgical management of nephrolithiasis in younger generation.

I do not think there are any problems with the method of study and the way the conclusion was drawn. However, I think there is a problem with the statistical method.

The size of the stones and the length of medical leave are continuous numbers, and you mention that you did T-tests in the Method section. However, since this is comparison between three groups (2007-2011, 2012-2016, and 2017-2021), you need to do ANOVA. If those numbers are not normally distributed, I think you need to use Kruskal-Wallis test. 

If you change the statistical method, I do not see any other major problem.

Only another thing, in the Conclusion section, you mention that endoscopic surgery will be the standard method for all ages in the near future, is it correct to assume the same for young children? I think you need a supplementary explanation.

Author Response

Reviewer #1

Thank you very much for agreeing to revise our manuscript and the time dedicated. 

The size of the stones and the length of medical leave are continuous numbers, and you mention that you did T-tests in the Method section. However, since this is comparison between three groups (2007-2011, 2012-2016, and 2017-2021), you need to do ANOVA. If those numbers are not normally distributed, I think you need to use Kruskal-Wallis test. If you change the statistical method, I do not see any other major problem.

We greatly appreciate this important comment.

In the METHODS section we have added a correction as follows                      (p.3 1st paragraph):

" Categorical variables were compared by Chi-Square or Fisher exact tests, while T-tests were used to examine differences in means of continuous variables. ANOVA test has been used to compare  normally distributed variables  whereas Kruskal-Wallis test has been implemented for the abnormally distributed ones."

 In the Conclusion section, you mention that endoscopic surgery will be the standard method for all ages in the near future, is it correct to assume the same for young children? I think you need a supplementary explanation.

This is already mentioned in the CONCLUSION paragraph (p.6 last paragraph):

" We predict that endoscopic surgery may soon become the new standard of care for most stones and for all age groups." (and children is no exception)

Reviewer 2 Report

Authors presented their study on the changing Trends in Surgical Management of Nephrolithiasis Among Young Adults. However, there was no addition to literature. Furthermore, authors discussed treatment of nephrolithiasis in children while the current study was on adults. 

There is also another important point that may lead to bias of results as there is a trend towards more definitive management in military community. SWL therapy usually requires multiple sessions with subsequent multiple leaves, so URS is preferred. Furthermore, SWL require follow-up for possibility of steinstrasse and ureteric obstruction. Frequent follow-up may be difficult during military service which is another potential reason for preferring more definitive URS. 

Authors reported in their conclusion that they predict that endoscopic surgery may soon become the new standard of care for most stones and for all age groups. However, endourology is already the standard of care. 

Authors did not present the number of patients in each period. 

Authors used t-test to compare 3 groups. 

Authors did not analyse different factors that may affect the choice of SWL vs URS (i.e., lower calyceal stones, medical comorbidities, chemical composition of stones, urinary tract abnormalitis, ureteric obstruction and impacted stones, etc)

Author Response

Reviewer  #2

Thank you very much for agreeing to revise our manuscript and the time dedicated. 

Authors presented their study on the changing Trends in Surgical Management of Nephrolithiasis Among Young Adults. However, there was no addition to literature.

We stated in the INTRODUCTION paragraph (p.2 3rd paragraph) that young adults

 " represent a population between children and adults that has not been studied in isolation"

and this is what makes the present paper innovative.

Authors discussed treatment of nephrolithiasis in children while the current study was on adults. 

Indeed, we discuss nephrolithiasis in children, but this comes all along in the context of treating young adults.

We start the DISCUSSION paragraph (p.4) with the current guidelines in nephrolithiasis,  turn to children and then state that (DISCUSSION 3rd. paragraph)

" Like the pediatric population with nephrolithiasis, young adults with nephrolithiasis usually are free of the associated comorbidities seen in adults and have similar relatively low body mass and fat content".

From here on, all we cite in the DISCUSSION paragraph regarding children is compared right away to our young adults (=the study) population.

There is also another important point that may lead to bias of results as there is a trend towards more definitive management in military community. SWL therapy usually requires multiple sessions with subsequent multiple leaves, so URS is preferred. Furthermore, SWL require follow-up for possibility of steinstrasse and ureteric obstruction. Frequent follow-up may be difficult during military service which is another potential reason for preferring more definitive URS. 

Thank you for the comment. In our cohort the military personnel have been treated by civilian urologists and their active treatment was according to the overall common practice with no regard to their military profession. Israel is a small country with a wide variety of medical facilities, hence, follow up protocols were not part of the decision making process. 

Authors reported in their conclusion that they predict that endoscopic surgery may soon become the new standard of care for most stones and for all age groups. However, endourology is already the standard of care.

We definitely  agree that endourology is the standard of care, but that's not what is said.

It is said , that endoscopic surgery will replace SWL in all age groups (which is not the current state worldwide). 

As also mentioned on the  DISCUSSION paragraph (p.5 6th paragraph):

 "Our results support the 2016 prediction of Heers et al. that "we are likely to see ureteroscopy overtake SWL for the first time in the next few years" "

Authors did not present the number of patients in each period. 

Thank you for the comment. Patients' numbers have been added and incorporated in the text.

RESULTS  paragraph p.3 2nd paragraph:

" Figure 1 depicts the changing trends in surgical management of nephrolithiasis during a 15-year period divided into three parts (number of patients in each time period): 2007-2021 (n=590), 2012-2016 (n=685), and 2017-2021 (n=580)."

Authors used t-test to compare 3 groups.

Thank you for the comment. We used Kruskal-Wallis test. The full description was added to the statistics section.

p.3 1st paragraph:

" Continuous variables are presented as median (interquartile range [IQR]), and categorical variables as percentages of individuals in each group. Categorical variables were compared by Chi-Square or Fisher exact tests, while T-tests were used to examine differences in means of continuous variables. ANOVA test has been used to compare  normally distributed variables  whereas Kruskal-Wallis test has been implemented for the abnormally distributed ones".   

Authors did not analyse different factors that may affect the choice of SWL vs URS (i.e., lower calyceal stones, medical comorbidities, chemical composition of stones, urinary tract abnormalitis, ureteric obstruction and impacted stones, etc).

Thank you for this comment. The scope of this study  was to highlight  the significant changes in surgical treatment modalities. Our cohort is of young healthy adults with no substantial comorbidities. We believe that the changes in stone location during the time periods is not a differential bias.

Reviewer 3 Report

I revised manuscript number jpm-1809538, titled “Changing Trends in Surgical Management of Nephrolithiasis Among Young Adults: A 15-year Population-Based Study”.

In this study, the authors to explore the trends in surgical management of nephrolithiasis in a large population of young adults.

This paper is very interesting; however, there are some specifications that need to be done.

Introduction

-       At the end of the introduction, you mention “…who represent a population between children and adults that has not been studied in isolation”. You already mention young adults, that line is redundant. You could mention that is a population not yet studied.

Methods

-       You should better specify your inclusion and exclusion criteria

Discussion

-       In your discussion you did not mention the relationship between the stone size and the election of the surgery. It would be accurate if you could mention that.

Author Response

Reviewer #3

Thank you very much for agreeing to revise our manuscript and for the time dedicated.

-       At the end of the introduction, you mention “…who represent a population between children and adults that has not been studied in isolation”. You already mention young adults, that line is redundant. You could mention that is a population not yet studied.

      In the INTRODUCTION  paragraph, this is the first time we refer young adults. Beforehand we mentioned adults, then children and finally we addressed this specific group that, in our modest opinion, needed to be meticulously defined.

Methods

-    You should better specify your inclusion and exclusion criteria

Thank you for this comment.

      We have added the following sentence at the end of MATERIALS AND METHODS/Study participants paragraph on p.2:

" For inclusion into this study, participants had to be 16-19 years of age at the time of their medical board examination".

Discussion

-       In your discussion you did not mention the relationship between the stone size and the election of the surgery. It would be accurate if you could mention that.

      This is a very good point, but you need to remember that we retrospectively analyzed a large military registry database of multiple medical centers and surgeons whom we were entirely blinded to. As we were not the surgeons ourselves, and we have no way to get to the real surgeons, we can't say anything regarding stone size and the selected surgery.

Round 2

Reviewer 2 Report

Authors tried to reply to some of the previous queries. I still have comments. 

- Authors mentioned that: [The purpose of the present study was to explore the trends in surgical management of nephrolithiasis in a large population of young adults, who represent a population between children and adults that has not been studied in isolation.]. However, there is no need to study this particular age group. Young adults are similar to adults and there will be no expected change between them. On the other hand, children differs due to the fact of smaller caliber ureters that may make ureteroscopic instrumentation difficult.   

- I previous reported in my comments that: [There is also another important point that may lead to bias of results as there is a trend towards more definitive management in military community. SWL therapy usually requires multiple sessions with subsequent multiple leaves, so URS is preferred. Furthermore, SWL require follow-up for possibility of steinstrasse and ureteric obstruction. Frequent follow-up may be difficult during military service which is another potential reason for preferring more definitive URS.] Authors replied by: [ In our cohort the military personnel have been treated by civilian urologists and their active treatment was according to the overall common practice with no regard to their military profession.] However, according to guidelines, there are always options. These options are considered common practice which is affected by social factors and patient choice. Thus, the decision process is affected directly or indirectly by the patient being in military service. Furthermore, these patients were already presenting with stones to military service and were not treated previously. Thus, one of the major factors that pushed towards treatment was the recruitment into military service. 

- The statistical paragraph is still incorrect. Authors added (Kruskal-Wallis test) according to my recommendation and other reviewers as the comparison was between 3 groups and T-test should not be used. However, authors still reporting t-test although there was no comparison between 2 groups. Furthermore, when sample size is large (here exceeding 500 in each group), parametric tests should be used. However, authors used non-parametric tests. 

- Authors reported: [The most recent (2016) American and European Urological Associations guidelines on nephrolithiasis recommend that, similar to adults, children should be offered SWL or Retrograde Intra-Renal Surgery (RIRS) for stones < 20mm]. However, 2016 is not a recent publication. EUA guidelines is published annually. 

- We commented previously: [Authors did not analyse different factors that may affect the choice of SWL vs URS (i.e., lower calyceal stones, medical comorbidities, chemical composition of stones, urinary tract abnormalitis, ureteric obstruction and impacted stones, etc).] Authors replied by: [Our cohort is of young healthy adults with no substantial comorbidities. We believe that the changes in stone location during the time periods is not a differential bias.]. However, the presence of abnormalities, certain chemical composition of stones, lower calyceal stones, associated ureteric obstruction or impacted stones are important factors that should be compared among the different 5 years periods. 

Author Response

Dear Reviewer,

First and foremost we would like to thank you again  for the time you dedicated to revise our manuscript. It is GREATLY appreciated. Below you will find our responses to your comments. We certainly agree with some of them, while with other we don't. You will find a detailed explanations to the points of controversy and we do hope that we managed to convince you. We also hope, that following all the corrections you will find our paper suitable for acceptance to the Journal of Personalized Medicine.

- Authors mentioned that: [The purpose of the present study was to explore the trends in surgical management of nephrolithiasis in a large population of young adults, who represent a population between children and adults that has not been studied in isolation.]. However, there is no need to study this particular age group. Young adults are similar to adults and there will be no expected change between them. On the other hand, children differs due to the fact of smaller caliber ureters that may make ureteroscopic instrumentation difficult.

We definitely do not agree with this determination,   and we  further addressed  this  point in our DISCUSSION  paragraph on pp.5-6:

"Like the pediatric population with nephrolithiasis, young adults with nephrolithiasis usually are free of the associated comorbidities seen in adults and have similar relatively low body mass and fat content.

We therefore would have expected to observe trends among them similar to those observed by Park et al among children [6]. …Opposite to our expectation, we evidenced a near-total disappearance of SWL".     

If we may, analyzing the results of certain procedure in various age groups and within a certain age group  is a common practice in the medical literature. The respectful reviewer may want to browse, for instance, the following publications:

  1. Thangavelu M, Sawant A, Sayed AA, et al: Retrograde intra renal surgery (RIRS) for upper urinary tract stones in children below 12 years of age: a single centre experience. Arch Ital Urol Androl 2022; 94(2): 190-4.
  2. Abu Ghazale LA, Shunaigat A, Budair Z: Retrograde intra renal lithotripsy for small stone in prepubertal children. Saudi J KidneyDis Transpl 2011; 22(3): 492-6.
  3. Islamoglu E, Tas S, Karamik K, et al: Does extracorporeal shock wave lithotripsy-related pain get affected by menstrual cycle and menopause ? Urolithiasis 2019; 47(6): 575-81    

- I previous reported in my comments that: [There is also another important point that may lead to bias of results as there is a trend towards more definitive management in military community. SWL therapy usually requires multiple sessions with subsequent multiple leaves, so URS is preferred. Furthermore, SWL require follow-up for possibility of steinstrasse and ureteric obstruction. Frequent follow-up may be difficult during military service which is another potential reason for preferring more definitive URS.] Authors replied by: [ In our cohort the military personnel have been treated by civilian urologists and their active treatment was according to the overall common practice with no regard to their military profession.] However, according to guidelines, there are always options. These options are considered common practice which is affected by social factors and patient choice. Thus, the decision process is affected directly or indirectly by the patient being in military service. Furthermore, these patients were already presenting with stones to military service and were not treated previously. Thus, one of the major factors that pushed towards treatment was the recruitment into military service. 

We are sorry, but your assumption is incorrect. In our country there is an excellent medical system that provides an excellent service to all citizens, both civilians and soldiers,  in a timely manner. Those candidates for military service would have definitely been taken care of even if they hadn't been recruited into military service. The recruitment to military service   facilitated in better recording the outcome of their medical treatment, and did not push towards any kind of treatment whatsoever.

- The statistical paragraph is still incorrect. Authors added (Kruskal-Wallis test) according to my recommendation and other reviewers as the comparison was between 3 groups and T-test should not be used. However, authors still reporting t-test although there was no comparison between 2 groups. Furthermore, when sample size is large (here exceeding 500 in each group), parametric tests should be used. However, authors used non-parametric tests. 

Thank you for your comments. We revised the data with our statistician.  We acknowledge that parametric tests have more power in group comparison, but our data is not equally distributed and descriptive in its nature, thus, Ksruskal –Wallis test was used. Your note regarding the parametric tests including the T test is correct and we deleted it from the text (p.3):

"Categorical variables were compared by Chi-Square or Fisher exact tests. Kruskal-Wallis was implemented for the comparison of multiple groups consisting of not-normally distributed variables.  Coefficients are presented as 95% confidence intervals (95% CI) and p-values".  

- Authors reported: [The most recent (2016) American and European Urological Associations guidelines on nephrolithiasis recommend that, similar to adults, children should be offered SWL or Retrograde Intra-Renal Surgery (RIRS) for stones < 20mm]. However, 2016 is not a recent publication. EUA guidelines is published annually.

Thank you for this comment, and we certainly apologize for this mistake . The most recent guidelines on nephrolithiasis were last updated in 2016 by the AUA. Indeed, the most recent EAU guidelines are dated 2022, though nothing has changed over the past 6 years… We have changed the text and the references accordingly:

p.4:

"The most recent American (2016) and European Urological Association (2022) guidelines on nephrolithiasis  recommend that, similar to adults, children should be offered  SWL or Retrograde Intra-Renal Surgery (RIRS) for stones < 20mm while stones > 20mm will preferably go to PCNL [14-15]" .

Ref. #14 (p.6): Skolarikos A, Neisius A, Petrik A, et al. EAU guidelines on  urolithiasis. https://uroweb.org/guidelines/urolithiasis

- We commented previously: [Authors did not analyse different factors that may affect the choice of SWL vs URS (i.e., lower calyceal stones, medical comorbidities, chemical composition of stones, urinary tract abnormalitis, ureteric obstruction and impacted stones, etc).] Authors replied by: [Our cohort is of young healthy adults with no substantial comorbidities. We believe that the changes in stone location during the time periods is not a differential bias.]. However, the presence of abnormalities, certain chemical composition of stones, lower calyceal stones, associated ureteric obstruction or impacted stones are important factors that should be compared among the different 5 years periods.   

This is a good point, but you need to remember that we are dealing with  a large military registry database consisting of administrative data obtained from multiple medical centers.    Given its nature, it does not (and cannot) consist all the data that you mentioned, as this is the nature of all administrative database registries.

You may also want to browse references #6,10,12 in the present paper as well as the following:

Jayram G, Matlaga B: Contemporary practice patterns associated with percutaneous nephrolithotomy among certifying urologists. J Endourol 2014; 28(11): 1304-7 

All the above are examples of similar researches in the field that are also lacking the factors that you mentioned:   the presence of abnormalities, certain chemical composition of stones, lower calyceal stones, associated ureteric obstruction or impacted stones